# Aerodynamic Performance and Neuromuscular Control in Patients with Unilateral Vocal Fold Paralysis

**DOI:** 10.3390/diagnostics12123124

**Published:** 2022-12-11

**Authors:** Wei-Li Chung, Kuo-Cheng Liu, Hsiu-Feng Chuang, Yi-An Lu, Hsueh-Yu Li, Alice M. K. Wong, Yu-Cheng Pei, Tuan-Jen Fang

**Affiliations:** 1Department of Physical Medicine and Rehabilitation, Chang Gung Memorial Hospital at Linkou, 5 Fushing St., Taoyuan 333, Taiwan; 2Department of Physical Medicine and Rehabilitation, New Taipei Municipal TuCheng Hospital (Built and Operated by Chang Gung Medical Foundation), 6, Sec. 2, Jincheng Rd., New Taipei City 236, Taiwan; 3Master of Science Degree Program in Innovation for Smart Medicine, Chang Gung University, 259 Wen-Hwa 1st Road, Taoyuan 333, Taiwan; 4Department of Otolaryngology, Chang Gung Memorial Hospital at Linkou, 5 Fushing St., Taoyuan 333, Taiwan; 5School of Medicine, Chang Gung University, 259 Wen-Hwa 1st Road, Taoyuan 333, Taiwan; 6Center of Vascularized Tissue Allograft, Chang Gung Memorial Hospital at Linkou, 5 Fushing St., Taoyuan 333, Taiwan

**Keywords:** unilateral vocal fold paralysis, laryngeal electromyography, aerodynamics

## Abstract

Unilateral vocal fold paralysis (UVFP) causes glottal incompetence and poor vocal efficiency. The influence of laryngeal neuromuscular control on aerodynamics in UVFP remains unclear. This study investigated the relationship between laryngeal muscle activities using quantitative laryngeal electromyography (LEMG) and aerodynamics in UVFP. This prospective study recruited patients with UVFP, and the diagnosis was confirmed with videolaryngostroboscopy and LEMG. The patient received aerodynamic assessment and LEMG of the thyroarytenoid-lateral cricoarytenoid (TA-LCA) muscle complex and the cricothyroid (CT) muscle. The relationship between quantitative LEMG and aerodynamic parameters was analyzed. A total of 134 UVFP patients without concurrent CT muscle involvement were enrolled. Compared with the normal side, the peak turn frequency of the lesioned side was lower in the TA-LCA (*p* < 0.001) and CT (*p* = 0.048) muscles. Stepwise linear regression revealed that the turn ratio of TA-LCA muscles was a robust factor in the decrease in peak expiratory airflow (β = −0.34, *p* = 0.036), mean airflow during voicing (β = −0.28, *p* = 0.014), and aerodynamic power (β = −0.42, *p* = 0.019), and an increase in aerodynamic efficiency (β = 27.91, *p* = 0.012). In addition, the turn ratio of CT muscles was a potent factor in inducing an increase in aerodynamic resistance (β = 14.93, *p* = 0.029). UVFP without CT involvement still showed suppression of CT muscles on the lesioned side, suggesting that neurological impairment of the TA-LCA complex could cause asymmetrical compensation of CT muscles, further impeding aerodynamics. The residual function of TA-LCA muscle complexes facilitates less air leakage and power dissipation, enhancing aerodynamic efficiency. On the other hand, the symmetrical compensation of the CT muscles improves aerodynamic resistance.

## 1. Introduction

Unilateral vocal fold paralysis (UVFP) causes glottal insufficiency and accounts for most neurogenic dysphonia. As shown in Figure 1, the thyroarytenoid–lateral cricoarytenoid (TA-LCA) muscle complex is innervated by the recurrent laryngeal nerve (RLN) and adducts the vocal fold (VF). The cricothyroid (CT) muscle is innervated by the external branch of the superior laryngeal nerve (eSLN) and controls VF tension but has minimal or no effect on the position of the VF [1]. The laryngeal acoustic output is determined by glottic posture, which is controlled by intrinsic laryngeal muscles, and aerodynamic force, which is governed by the respiratory system and glottal resistance. UVFP manifests with effortful talking, vocal fatigue, and poor vocal efficiency, indicating the need for an exaggerated compensatory aerodynamic mechanism to produce an adequate acoustic output [2,3].

Both laryngeal electromyography (LEMG) and aerodynamic analysis are important tools that can yield cardinal information regarding the mechanisms of UVFP-induced dysphonia. The neuromuscular function of intrinsic laryngeal muscles can be assessed by LEMG. The conventional LEMG is used to determine the existence of denervation, but its assessment for motor unit recruitment is only semiquantitative [4,5]. In contrast, quantitative LEMG could be used to objectively measure the interference pattern in laryngeal muscles and can thus fathom the severity of neuromuscular impairment [6,7]. In addition, quantitative LEMG has been shown to correlate with volitional activity [8] and yield prognostic values in UVFP [9].

On the other hand, aerodynamic analysis is used to measure aerodynamic performance by quantifying parameters such as sound pressure, air pressure, and expiratory airflow. Via aerodynamic analysis, a lower laryngeal resistance, higher translaryngeal airflow, increased intraoral pressure, and increased phonation threshold pressure were observed in UVFP patients, a finding that could be accounted for by their VF incompetence [3,10,11]. However, the degree to which aerodynamic differences are attributed to the level of neuromuscular impairment remains unclear. Dewan et al. [12] investigated the relationship between laryngeal muscle activities and aerodynamics in UVFP using an animal model and found that, at the beginning of phonation, RLN activities that adduct the VF are necessary to induce a decrease in airflow. Nevertheless, SLN activities have a more sophisticated effect on phonation, as they affect phonatory onset pressure and airflow by modulating the tension and relative position of VF. SLN activities affecting aerodynamics may further interact with RLN activities through different strain levels and postures of VF [12,13].

To the best of our knowledge, few studies have yet characterized the correspondence between neuromuscular activities in laryngeal muscles and aerodynamics during voicing in patients with UVFP. Bielamowicz et al. [14] found that motor unit recruitment assessed by conventional LEMG significantly affects aerodynamics such as phonation time and airflow in UFVP patients. Decreased maximal phonation time and increased translaryngeal airflow were found in patients with reduced motor unit recruitment, but their recruitment analysis only applied a subjective grading system using 1+ to 4+ to assess the neuromuscular function of the intrinsic laryngeal muscles. In addition, LEMG findings in laryngeal muscles of different primary phonatory functions were combined to represent the whole laryngeal muscle tone. To comprehensively investigate the relationship between laryngeal muscle activities and voice aerodynamics in UVFP, the present study performed quantitative LEMG in each of the TA-LCA and CT muscles. Most importantly, bilateral laryngeal muscles were evaluated so that the functional or compensatory changes on the normal side could be elucidated. We hypothesized that the residual function of TA-LCA muscle complexes and the level of compensatory activities of the CT muscles could affect the change in aerodynamics [15].

## 2. Materials and Methods

### 2.1. Patients

The present prospective study recruited patients diagnosed with unilateral vocal fold paralysis (UVFP) from otolaryngology outpatient clinics at a single medical center from 1 October 2015 to 28 February 2019. The exclusion criteria were prior history of vocal fold paralysis, not cooperating with or refusing LEMG assessments, history of interventions to paralyzed vocal folds (such as laryngoplasty, intracordal injection, or laryngeal framework surgery), normal findings in TA-LCA from LEMG, or concurrent involvement of the CT muscle. Specifically, patients with abnormal findings in CT muscle were excluded because it has been shown that CT muscle involvement alone is an important factor in determining voice frequency [16,17], voice quality [18], and aerodynamic performance [12,19]. The study was approved by the Institutional Review Board of Chang Gung Medical Foundation, and informed consent was obtained from each participant before recruitment.

### 2.2. Assessments

All participants received assessments including videolaryngostroboscopy, LEMG, and aerodynamic evaluations. UVFP was confirmed by abnormal findings in both LEMG and videolaryngostroboscopy.

#### 2.2.1. Laryngeal Electromyography (LEMG)

An otolaryngologist and a physiatrist performed LEMG according to the standard protocol described as follows. Each participant sat on a chair specially designed for LEMG examination, with the neck extended and head supported by the adjustable neck–head rest. The LEMG activity of bilateral TA-LCA muscle complexes and cricothyroid muscles was recorded through the concentric needle electrode using the Nicolet Viking Select system (Cardinal Health, Dublin, OH, USA) with a ground-surface electrode adhered to the forehead. For the TA-LCA muscle complex, the concentric needle electrode was angled 15° superiorly and 30° laterally through off-midline insertion into the cricothyroid membrane. For the CT muscle, the electrode was inserted 5 mm away from the midline to the level of the center of cricothyroid membrane, with an angle of 50° laterally [20]. The bandpass filter was set between 20 Hz and 10 kHz. Before needle insertion, the overlying skin was anesthetized with 1 to 2 mL of 2% lidocaine hydrochloride. To examine the TA-LCA muscle complex, the participant was asked to sequentially produce three series of /e/ sounds at three voice intensities (soft, moderate, and loudest possible). Each /e/ sound lasted for at least 400 ms, with each inter-/e/ interval lasting over 200 ms [6]. To evaluate the cricothyroid muscle, the participant was asked to produce the three upward glissando /e/s at a comfortable voice intensity, with intervals lasting over 3 s [7]. The formal LEMG consists of insertional and spontaneous activities, a semiquantitative motor unit, and recruitment analyses. Pathological changes in LEMG are defined as the existence of spontaneous pathological activities (such as fibrillation, positive sharp waves, and complex repetitive discharge), >30% polyphasic waves, and decreased interference patterns.

#### 2.2.2. Quantitative LEMG Analysis

The raw LEMG data were first binned into nonoverlapping epochs, in which the TA-LCA muscle complex and the cricothyroid muscle were divided every 200 ms [6] and 50 ms [7], respectively. An automatic algorithm was used to find the timing of each turn, defined by a transition in the waveform polarity with an amplitude difference of at least 100 μV before and after the charge to exclude noise-related peaks, as depicted in the example reported in Figure 2. The turn frequency of each epoch was calculated by dividing the number of turns by the epoch duration. The peak turn frequency of each muscle was computed by averaging the highest three values of turn frequency, and the turn ratio was the ratio of the peak turn frequency of the lesioned side to that of the normal side, as shown in Equation (1):Turn ratio = (peak turn frequency of lesioned side)⁄(peak turn frequency of normal side)(1)

Therefore, the TA-LCA turn ratio is defined as the ratio of the peak turn frequency of the lesioned side of the TA-LCA complex to that of the normal side, while the CT turn ratio is the ratio of the peak turn frequency of the lesioned side of the CT muscle to that of the normal side.

#### 2.2.3. Videolaryngostroboscopy

Each patient was asked to vocalize an /e/ sound at habitual pitch and volume, while vocal fold vibration and glottis movement were visualized and recorded by videolaryngostroboscopy.

#### 2.2.4. Aerodynamic Analysis

Aerodynamic analysis for the voice was performed using the voicing efficiency protocol of the Phonatory Aerodynamic System (PAS) Model 6600 device (KayPENTAX Corp, Lincoln Park, NJ, USA) (see Figure 3A). Each participant was instructed to produce successive /pa-pa-pa/ utterances three times at habitual pitch and volume while placing the mask firmly on the face with the oral pressure tube inserted into the mouth without touching the tongue (see Figure 3B). Each participant was trained to produce the syllable sequence at a pace of 1.5–2 syllables per second to yield a stable pressure trace and equal rhythm. A certified speech pathologist conducted the aerodynamic analysis, and all parameters were measured in each of the three trials. The mean value of each parameter is presented.

The sound-pressure level (SPL) and mean airflow during voicing were measured from the voiced segments of the vowel /a/. Intra-oral peak air pressure during the production of the plosive consonant /p/ was measured to approximate subglottal pressure. The second to fifth peak intraoral pressures were averaged for each trial as the central and stable syllable repetitions were used in previous studies [19,21]. Mean SPL and mean airflow during voicing, along with peak air pressure, were then used by the PAS software to calculate aerodynamic power, resistance, and efficiency values using the following equations [22]:Aerodynamic power = mean peak air pressure × target airflow × 0.09806(2)
Aerodynamic resistance = mean peak air pressure/target airflow(3)
Aerodynamic efficiency = acoustic power/aerodynamic power(4)
Acoustic power = 1.4137 × 10^−7^ × 10^MEADB/10^(5)
where MEADB is the mean sound-pressure level taken from the sound-pressure contour regions that were used to establish the start and end locations for extracting the target airflow [22].

### 2.3. Statistical Analysis

All data were analyzed using SPSS software, version 20.0 (SPSS Inc., Chicago, IL, USA) and presented as the mean (standard deviation). A one-sample *t*-test was used to compare the aerodynamic data vs. the normative data of the age- and gender-matched healthy subjects [23]. A paired-samples *t*-test was used to compare quantitative LEMG parameters between the normal and lesioned sides. Pearson correlation was applied to assess the relationship between quantitative LEMG and aerodynamic parameters. Stepwise linear regression was used to identify the predictors of the aerodynamic parameters out of the candidate terms, including the TA-LCA turn ratio, CT turn ratio, and interaction of the TA-LCA and CT ratios. Statistical significance was defined as *p* < 0.05.

## 3. Results

### 3.1. Patient Characteristics

A total of 194 patients with UVFP were recruited, among whom 5 were excluded due to no TA-LCA involvement, 18 due to incomplete LEMG data, and 37 due to the coexistence of CT muscle involvement confirmed by LEMG (see Figure 4). The demographic data of the remaining 134 patients ((mean ± SD) age, (54.7 ± 12.9) years; 80 (59.7%) male) are listed in Table 1.

### 3.2. Aerodynamic Analysis

Comparison of aerodynamic parameters with age- and gender-matched healthy subjects showed that our patients had increased translaryngeal airflow, intraoral pressure, and aerodynamic power and decreased aerodynamic efficiency, a finding that was in line with an increase in phonatory effort in UVFP (Table 2) [3,10,11].

### 3.3. Quantitative Laryngeal Electromyography Assessment

Comparison of quantitative LEMG parameters between the normal and lesioned sides showed that, in TA-LCA muscle complexes, peak turn frequency in the normal side was remarkably higher than that in the lesioned side (1011.84 ± 371.37 vs. 348.39 ± 251.87 turn/s, *p* < 0.001). Surprisingly, the peak turn frequency in the CT muscle was lower in the lesioned side than in the normal side (862.08 ± 302.02 vs. 802.98 ± 298.77 turn/s, *p* = 0.048), suggesting that denervation to the TA-LCA muscle complex induced a minor decrease in the functional activities of the CT muscle on the same side (Table 3).

### 3.4. Correlations between Quantitative LEMG and Aerodynamic Parameters

Table 4 shows the correlation coefficients between parameters of quantitative LEMG and aerodynamics. Peak turn frequency of the normal-side TA-LCA muscle complex showed positive correlations with mean SPL during voicing (r = 0.17; *p* = 0.045), mean peak air pressure (r = 0.19; *p* = 0.025), peak expiratory airflow (r = 0.18; *p* = 0.041), and aerodynamic power (r = 0.24; *p* = 0.006) (see Figure 5(A1–A4)); peak turn frequency of the lesioned-side TA-LCA muscle complex showed positive correlations with maximal SPL (r = 0.17; *p* = 0.047) and aerodynamic efficiency (r = 0.26; *p* = 0.002) but negative correlations with mean airflow during voicing (r = −0.23; *p* = 0.008) and aerodynamic power (r = −0.18; *p* = 0.039) (see Figure 5(B1–B4)). These findings indicated that (1) the TA-LCA muscle complex was correlated with all of the aerodynamic parameters, except for the aerodynamic resistance, while, interestingly, (2) the TA-LCA muscle complexes in the normal and lesioned sides had opposite roles in terms of their correlation with translaryngeal airflow and aerodynamic power. Furthermore, the turn ratio of TA-LCA muscle complexes showed positive correlations with aerodynamic efficiency (r = 0.22; *p* = 0.012) but negative correlations with peak expiratory airflow (r = −0.18; *p* = 0.036), mean airflow during voicing (r = −0.21; *p* = 0.014), and aerodynamic power (r = −0.20; *p* = 0.019) indicating that the residual function of the TA-LCA muscle complexes was correlated with less air leakage during voicing, decreased aerodynamic power, and increased aerodynamic efficiency (see Figure 5(C1–C4)).

For the CT muscle, the peak turn frequency of the normal-side CT showed positive correlations with maximal SPL (r = 0.21; *p* = 0.015), mean SPL during voicing (r = 0.22; *p* = 0.009), and peak air pressure (r = 0.17; *p* = 0.047), while the peak turn frequency of the lesioned-side CT showed a positive correlation with maximal SPL (r = 0.17; *p* = 0.047), indicating that the CT muscle was mainly associated with SPL. Only the normal-side CT was further correlated with peak air pressure level. In addition, the turn ratio of CT muscles showed a positive correlation with aerodynamic resistance (r = 0.19; *p* = 0.029), indicating that the symmetric activation of bilateral CT muscles may predict an increase in aerodynamic resistance (see Figure 5(D1–D5)).

### 3.5. Prognostic Indicators of the Aerodynamic Parameters

Stepwise regression analysis, using turn ratios of TA-LCA muscle complexes and CT muscles and their interaction term as dependent variables and aerodynamic parameters as independent variables (Table 5), showed that turn ratio of TA-LCA muscle complexes was associated with peak expiratory airflow (β = −0.34; 95% CI, −065 to −0.02; *p* = 0.036), mean airflow during voicing (β = −0.28; 95% CI, −0.50 to −0.06; *p* = 0.014), aerodynamic power (β = −0.42; 95% CI, −0.78 to −0.07; *p* = 0.019), and aerodynamic efficiency (β = 27.91; 95% CI, 6.24 to 49.51; *p* = 0.012), indicating that residual function of the TA-LCA muscle complexes was a robust factor in inducing the decrease in air leakage and power dissipation during voicing and an increase in aerodynamic efficiency. On the other hand, the turn ratio of CT muscles could predict aerodynamic resistance (β = 14.93; 95% CI, 1.54 to 28.31; *p* = 0.029), indicating that aerodynamic resistance was influenced by the symmetric activation of the CT muscles but not by the TA-LCA muscle complex or either side of the CT muscles. Finally, the interaction between the turn ratio of TA-LCA muscle complexes and CT muscles did not predict any aerodynamic parameters.

## 4. Discussion

To the best of our knowledge, the present study is the first to evaluate the degree to which the maximal recruitment of each TA-LCA complex and CT muscle affects aerodynamics in patients with UVFP. Most importantly, the aforementioned analyses are based on the quantitative LEMG recorded in the normal side, lesioned side, as well as the relative difference between the normal and lesioned sides. The results showed that effects on aerodynamics affected by the maximal recruitment of TA-LCA differ between the normal and lesioned sides. Specifically, a higher maximal recruitment in the lesioned side is associated with less air leakage during voicing, alleviated aerodynamic power, and increased aerodynamic efficiency; in contrast, a higher maximal recruitment in the normal side is associated with increased translaryngeal airflow and air pressure combined with increased aerodynamic power. Furthermore, the relative maximal recruitment of TA-LCA muscle complexes facilitates voicing efficiency by decreasing the translaryngeal airflow required for phonation. The same analyses between CT and aerodynamics showed a completely different pattern by demonstrating that the symmetricity of the CT muscle activities improved laryngeal resistance in UVFP patients, indicating that denervation to the TA-LCA muscle can cause a functional imbalance of CT muscles and that their effects on laryngeal resistance differ between normal and lesioned TA-LCA.

The TA-LCA muscle complex primarily facilitates voicing by adducting the VF to decrease the glottal gap, resulting in decreased translaryngeal air flow and subglottal pressure combined to produce a lesser aerodynamic power required to initiate and maintain phonation [24]. As seen in the correlation between LEMG in the lesioned-side TA-LCA and aerodynamics, the more severe the denervation, the more translaryngeal air flow due to the widening glottal gap and the resultant increased aerodynamic power are required for phonation [12], together causing breathy and effortful voicing. Hence, the recovery or medial vocal fold medialization procedures of the lesioned-side TA-LCA primarily assumed an energy-conserving mechanism to improve voicing by decreasing air leakage.

In contrast, the maximal recruitment in the normal-side TA-LCA is associated with increased translaryngeal airflow and air pressure combined with increased aerodynamic power but not with aerodynamic efficiency, indicating less conversion efficiency by enhancing subglottal pressure and the translaryngeal airflow required for phonation. Several speculations may account for this energy-consuming mechanism. First, Dewan et al. [12] and Azar and Chhetri [13] found that, as the activation of TA-LCA increased, the prephonatory glottal gap closed gradually, resulting in a decrease in the demand for airflow to phonation. However, once achieving the peak activation of TA-LCA, the overall stiffness of VF increased [13], and the position of VF would be at hyperadduction, causing the height of bilateral VFs to be mismatched and leading to poor phonatory posture [12], which requires more aerodynamic energy (pressure and flow) to achieve phonation. Second, supraglottal structures, such as false vocal folds, may be recruited to compensate for underlying glottal inadequacy in patients with UVFP [25]. The false VF in the normal-side TA-LCA may be at a position of hyperadduction during phonation in UVFP patients [26]. The sphincter-like contraction of false VF further caused higher subglottic pressure and translaryngeal airflow [27]. Third, Chhetri and Neubauer investigated the different roles of thyroarytenoid (TA) and lateral cricoarytenoid (LCA) muscles in phonation and found that, as LCA adducted the cartilaginous VF to close the posterior glottis and constantly reduced the translaryngeal airflow requirement, TA first adducted and then shortened the membranous VF to increase the translaryngeal airflow. Thus, TA and LCA were antagonistic for translaryngeal airflow, especially at a higher level of TA-LCA activation [28]. The peak turn frequency of the normal-side TA-LCA in this study (1011.84 ± 371.37 turn/s) was higher than that of the healthy subjects in Statham’s [6] work (450 turn/s), suggesting a compensatory mechanism. Chen et al. [29] also reported similar findings where the peak turn frequency (582 ± 212 turn/s) of the opposite normal mobile vocal fold in UVFP patients was higher compared to Statham et al. [6]; they speculated that this was possibly attributable to the difference in age and gender, although compensation was possible if the opposite normal mobile vocal fold crossed the midline (hyperadducted). Our study shed light on the influence of the residual function of the TA-LCA muscle complexes to facilitate less air leakage during voicing, lower aerodynamic power, and increased aerodynamic efficiency.

The CT muscle elongates the vocal fold, stiffens the cover layer, and is primarily regarded as the controller of vocal range and glottal vibration [12,16,17,18,19,24]. There is increasing evidence suggesting its role in aerodynamics such as SPL [20], phonation onset pressure, phonation onset flow [12,24], and vocal efficiency [30], and its effects on aerodynamics may be further influenced by the activation level of TA-LCA and prephonatory VF posture. For the paralyzed TA-LCA muscle complex at the lesioned site (i.e., flaccid and abducted VF), the CT muscle tenses the VF and causes lengthening of the anteroposterior diameter of the glottis with an increased degree of adduction, which further resists the lateralizing airflow during voicing to compensate for ipsilateral RLN paralysis (compensatory falsetto) [12,31]. For the moderately contracted TA-LCA (i.e., stiff and adducted VF), the CT muscle further increases VF stain, counteracts the medial bulging effects of the TA muscle, and increases cartilaginous glottal width, together with increasing VF tension and glottal gap, requiring more subglottal pressure and airflow for phonation [12,32]. Surprisingly, the peak turn frequency of CT on the normal side was higher than that on the lesioned side, reflecting the asymmetric function of bilateral CT muscles in UVFP patients, despite no neuromuscular impairment of bilateral CT muscles. According to Chhetri et al., there is an antagonistic interaction between laryngeal adductors and the CT muscle on the VF strain [32]; hence, the higher activity of the CT muscle at the normal site may be derived from ipsilateral vigorous TA-LCA muscle complex activity. In addition, due to an antagonistic relationship, a functional decrease in the lesioned-side CT muscle activities could be induced by denervation of the TA-LCA muscle complex. Decreased glottal resistance was another prominent feature of UVFP, and our study further found that the CT muscles robustly affected the aerodynamic resistance in UVFP. Specifically, it was the symmetrical activation of CT muscles that enhanced laryngeal resistance in UVFP. Hence, the neurological impairment of the TA-LCA complex could cause functional asymmetry of CT muscles, further impeding phonatory control in patients with UVFP.

## 5. Limitation

The present study has several limitations. First, respiratory function, which drives the aerodynamics of voice, was not measured in the present study. Second, the patients in the current case series differed in their etiologies and severities, so subgroup analysis will be applied when a larger sample size can be achieved in the future. Third, there was a wide range of time post-paralysis (0.5–249.7 months), which could affect clinical symptoms and LEMG findings because reinnervation might happen over several months after nerve injury [9]. Further studies are needed to analyze the course of reinnervation and its influence on aerodynamics in patients with UVFP.

## 6. Conclusions

The present study found an association between neuromuscular function of intrinsic laryngeal muscles and aerodynamics, which sheds light on the mechanisms underlying neuromuscular control of vocal efficiency and provides a basis for therapeutic applications to facilitate phonatory efficiency. Specifically, the residual function of the TA-LCA muscle complexes facilitates less air leakage and alleviates voice effort, enhancing aerodynamic efficiency. Symmetrical activities of bilateral CT muscles improve aerodynamic resistance, implying that neurological impairment of the TA-LCA complex could cause asymmetrical compensation of CT muscles, further impeding aerodynamics. Therefore, the improved residual function and symmetrical neuromuscular controls of TA-LCA muscle complexes and CT muscles are the key elements of efficient energy conversion in phonation in patients with UVFP.

## Figures and Tables

**Figure 1 diagnostics-12-03124-f001:**
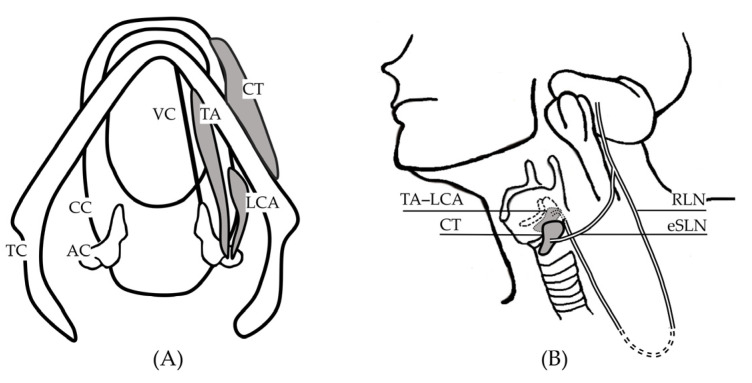
Schematic diagrams of (**A**) superior view with laryngeal muscles and (**B**) innervation of laryngeal muscles. TC, thyroid cartilage; CC, cricoid cartilage; AC, arytenoid cartilage; VC, vocal cord; TA, thyroarytenoid muscle; LCA, lateral cricoarytenoid muscle; CT, cricothyroid muscle; RLN, recurrent laryngeal nerve; eSLN, external branch of the superior laryngeal nerve.

**Figure 2 diagnostics-12-03124-f002:**
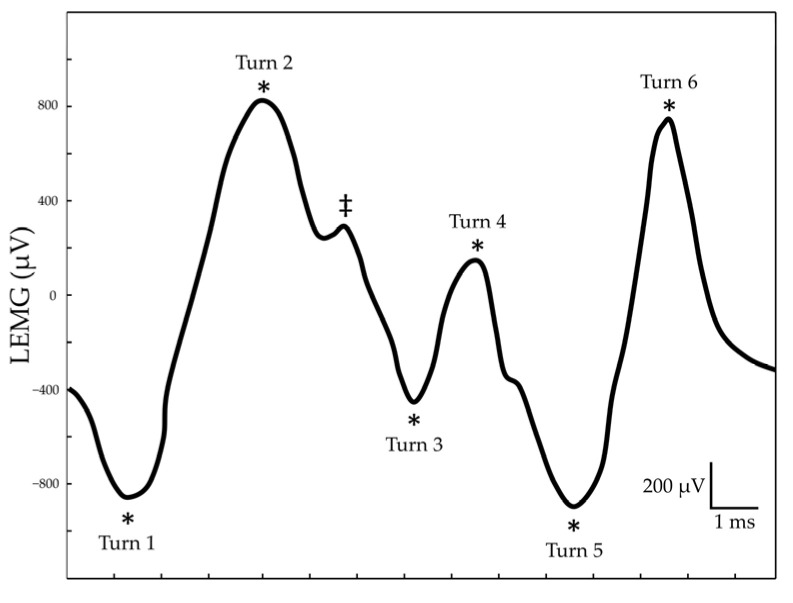
Example of a turns/s calculation in a 15 ms LEMG segment. A ‘‘turn’’ occurs at each peak of the measured signal (asterisks). The total number of detected turns was six, as the peak observed between Turn 2 and Turn 3 (double dagger) was below the threshold of 100 µV and therefore excluded from calculation.

**Figure 3 diagnostics-12-03124-f003:**
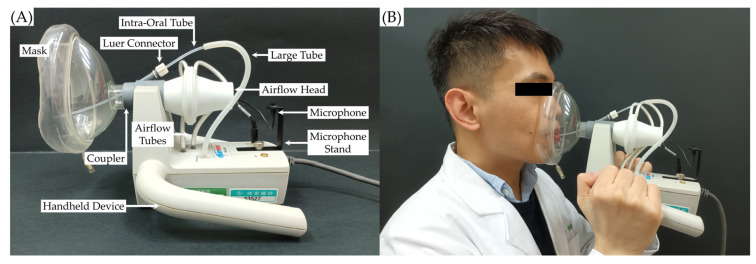
The Phonatory Aerodynamic System external module. (**A**) The coupler is attached to the airflow head (pneumotach) and the intra-oral pressure kit is attached to the handheld device; the glottal airflow rate is estimated from the oral airflow rate during vowel production and the subglottal air pressure is estimated from the intraoral air pressure produced during stop consonant production. The microphone is placed in its stand; the acoustic signal is picked up by a microphone and the estimate of sound-pressure level (SPL) is taken from the acoustic signal at the same time points in each vowel so that airflow measurements are obtained to facilitate the interpretation of airflow and air pressure measures. (**B**) The correct way to hold the Phonatory Aerodynamic System external module during data capture and the signals from airflow, air pressure, and microphone systems are acquired simultaneously during phonatory tasks.

**Figure 4 diagnostics-12-03124-f004:**
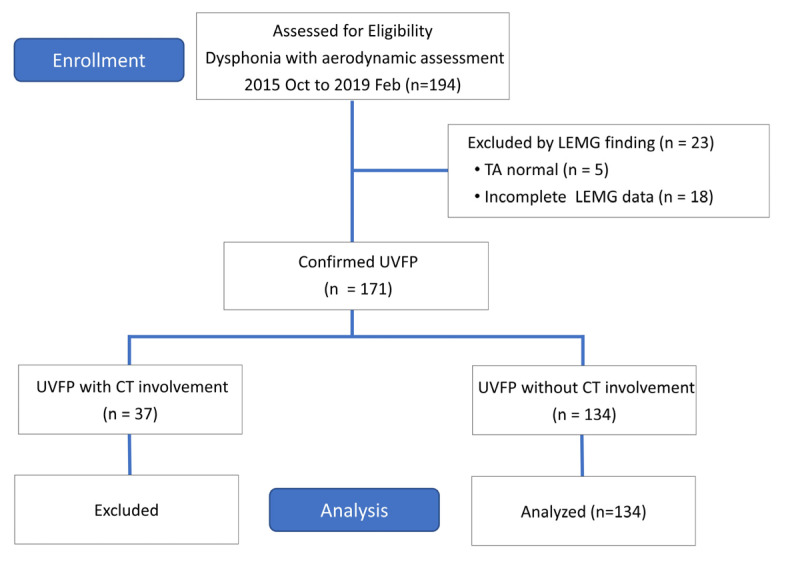
Experimental flowchart. A total of 194 patients with UVFP were recruited, among whom 5 were excluded due to no TA-LCA involvement, 18 due to incomplete LEMG data, and 37 due to the coexistence of CT muscle involvement.

**Figure 5 diagnostics-12-03124-f005:**
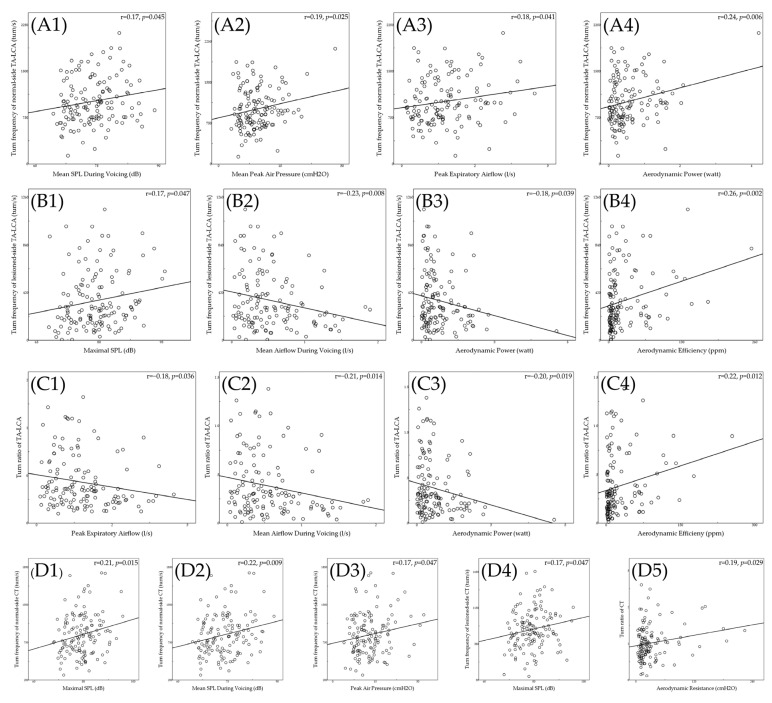
Scatter plots showing the relationship between parameters of quantitative LEMG and aerodynamics. Peak turn frequency of the normal-side TA-LCA muscle complex showed significant positive correlations with (**A1**) mean SPL during voicing (r = 0.17; *p* = 0.045), (**A2**) mean peak air pressure (r = 0.19; *p* = 0.025), (**A3**) peak expiratory airflow (r = 0.18; *p* = 0.041), and (**A4**) aerodynamic power (r = 0.24; *p* = 0.006). Peak turn frequency of the lesioned-side TA-LCA muscle complex showed significant positive correlations with (**B1**) maximal SPL (r = 0.17; *p* = 0.047) and (**B4**) aerodynamic efficiency (r = 0.26; *p* = 0.002) but significant negative correlations with (**B2**) mean airflow during voicing (r = −0.23; *p* = 0.008) and (**B3**) aerodynamic power (r = −0.18; *p* = 0.039). The turn ratio of TA-LCA muscle complexes showed significant negative correlations with (**C1**) peak expiratory airflow (r = −0.18; *p* = 0.036), (**C2**) mean airflow during voicing (r = −0.21; *p* = 0.014), and (**C3**) aerodynamic power (r = −0.20; *p* = 0.019) but significant positive correlations with (**C4**) aerodynamic efficiency (r = 0.22; *p* = 0.012). The peak turn frequency of the normal-side CT showed significant positive correlations with (**D1**) maximal SPL (r = 0.21; *p* = 0.015), (**D2**) mean SPL during voicing (r = 0.22; *p* = 0.009), and (**D3**) peak air pressure (r = 0.17; *p* = 0.047). The peak turn frequency of the lesioned-side CT showed a significant positive correlation with (**D4**) maximal SPL (r = 0.17; *p* = 0.047). The turn ratio of CT muscles showed a significant positive correlation with (**D5**) aerodynamic resistance (r = 0.19; *p* = 0.029). TA-LCA, thyroarytenoid–lateral cricoarytenoid; CT, cricothyroid; SPL, sound-pressure level.

**Table 1 diagnostics-12-03124-t001:** The demographics of the UVFP patients without cricothyroid involvement.

	N = 134		
Age (year)	54.7	±	12.9
Sex			
Male	80		(59.7%)
Female	54		(40.3%)
Paralysis side			
Left	96		(71.6%)
Right	38		(28.4%)
Time post-paralysis (month)	5	±	21.9
Pathogenesis (number, %)			
Thoracic surgery			
Esophageal	24		(17.9%)
Mediastinum surgery	17		(12.7%)
Heart surgery	13		(9.7%)
Lung surgery	11		(8.2%)
Total	65		(48.5%)
Other etiology			
Thyroidectomy	37		(27.6%)
Idiopathic	18		(13.4%)
Skull base or brain surgery	3		(2.2%)
Cervical spine surgery	7		(5.2%)
Thyroid tumor	2		(1.5%)
Other	2		(1.5%)
Total	69		(51.5%)

Data are presented as mean ± standard deviation or as number of patients. N, total number.

**Table 2 diagnostics-12-03124-t002:** Comparison of aerodynamic parameters with the normative data for the age-and gender-matched subjects.

Aerodynamic Parameters		Sample	Normative Data [23]	*p*-Value ^a^
Maximal SPL	N = 134	79.23 ± 5.85		
(dB)	M = 80	81.39 ± 5.67	82.02 ± 3.27	0.321
	F = 54	76.04 ± 4.53	79.91 ± 4.56	<0.001 ***
Mean SPL During Voicing	N = 134	74.02 ± 5.46		
(dB)	M = 80	76.00 ± 5.36	78.64 ± 2.99	<0.001 ***
	F = 54	71.09 ± 4.17	75.68 ± 4.24	<0.001 ***
Peak Air Pressure	N = 134	13.11 ± 5.44		
(cmH_2_O)	M = 80	14.42 ± 5.69	8.64 ± 2.23	<0.001 ***
	F = 54	11.17 ± 4.42	7.49 ± 2.64	<0.001 ***
Mean Peak Air Pressure	N = 134	9.46 ± 4.15		
(cmH_2_O)	M = 80	10.34 ± 4.37	6.91 ± 1.68	<0.001 ***
	F = 54	8.17 ± 3.44	5.76 ± 1.51	<0.001 ***
Peak Expiratory Airflow	N = 134	0.90 ± 0.57		
(L/s)	M = 80	1.17 ± 0.56	0.32 ± 0.15	<0.001 ***
	F = 54	0.48 ± 0.26	0.22 ± 0.10	<0.001 ***
Mean Airflow During Voicing	N = 134	0.55 ± 0.41		
(L/s)	M = 80	0.72 ± 0.42	0.19 ± 0.10	<0.001 ***
	F = 54	0.29 ± 0.19	0.13 ± 0.06	<0.001 ***
Aerodynamic Power	N = 134	0.61 ± 0.64		
(watt)	M = 80	0.84 ± 0.72	0.13 ± 0.07	<0.001 ***
	F = 54	0.27 ± 0.28	0.08 ± 0.05	<0.001 ***
Aerodynamic Resistance	N = 134	31.77 ± 36.62		
(cmH_2_O)	M = 80	23.75 ± 33.58	45.30 ± 26.16	<0.001 ***
	F = 54	43.67 ± 37.98	48.50 ± 26.28	0.354
Aerodynamic Efficiency	N = 134	26.70 ± 39.65		
(ppm)	M = 80	24.96 ± 40.33	60.80 ± 37.79	<0.001 ***
	F = 54	29.28 ± 38.84	59.84 ± 24.81	<0.001 ***

Data are presented as mean ± standard deviation. ^a^ One-sample *t*-test. SPL, sound-pressure level. *** *p* < 0.001. N, total number; M, male; F, female.

**Table 3 diagnostics-12-03124-t003:** Comparison of quantitative LEMG between the normal side and lesioned side.

Recruitment Analysis (N = 134)	Normal Side	Lesioned Side	*p*-Value ^a^
Peak frequency of TA-LCA (turn/s)	1011.84 ± 371.37	348.39 ± 251.87	<0.001 ***
Turn ratio of TA-LCA	0.38	±0.31	
Peak frequency of CT (turn/s)	862.08 ± 302.02	802.98 ± 298.77	0048 *
Turn ratio of CT	1.02	±0.46	

Data are presented as mean ± standard deviation. ^a^ Paired-samples *t*-test. * *p* < 0.05; *** *p* < 0.001. LEMG, laryngeal electromyography; TA-LCA, thyroarytenoid–lateral cricoarytenoid; CT, cricothyroid; N, total number.

**Table 4 diagnostics-12-03124-t004:** Correlation between LEMG and aerodynamics.

		Turn Frequency of Normal-Side TA-LCA (Turn/s)	Turn Frequency of Lesioned-Side TA-LCA (Turn/s)	Turn Ratio of TA-LCA	Turn Frequency of Normal-Side CT (Turn/s)	Turn Frequency of Lesioned-Side CT (Turn/s)	Turn Ratio of CT
Maximal SPL (dB)	r	0.14	0.17	0.15	0.21	0.17	−0.02
	*p* ^a^	0.097	0.047 *	0.077	0.015 *	0.047 *	0.792
Mean SPL During Voicing (dB)	r	0.17	0.14	0.11	0.22	0.15	−0.07
	*p*	0.045 *	0.099	0.209	0.009 **	0.093	0.445
Peak Air Pressure (cmH_2_O)	r	0.15	0.01	−0.03	0.17	0.14	−0.05
	*p*	0.079	0.882	0.767	0.047 *	0.112	0.544
Mean Peak Air Pressure (cmH_2_O)	r	0.19	−0.01	−0.05	0.17	0.13	−0.04
	*p*	0.025 *	0.945	0.545	0.053	0.122	0.621
Peak Expiratory Airflow (L/s)	r	0.18	−0.16	−0.18	0.13	0.03	−0.13
	*p*	0.041 *	0.065	0.036 *	0.13	0.721	0.136
Mean Airflow During Voicing (L/s)	r	0.14	−0.23	−0.21	0.09	−0.003	−0.10
	*p*	0.114	0.008 **	0.014 *	0.315	0.971	0.245
Aerodynamic Power (watt)	r	0.24	−0.18	−0.20	0.15	0.06	−0.10
	*p*	0.006 **	0.039 *	0.019 *	0.09	0.473	0.274
Aerodynamic Resistance (cmH_2_O)	r	−0.15	0.08	0.10	−0.15	0.05	0.19
	*p*	0.091	0.355	0.253	0.087	0.606	0.029 *
Aerodynamic Efficiency (ppm)	r	−0.01	0.26	0.22	0.07	0.06	−0.01
	*p*	0.928	0.002 **	0.012 *	0.430	0.489	0.956

^a^ Pearson correlation coefficient. * *p* < 0.05; ** *p* < 0.01. LEMG, laryngeal electromyography; TA-LCA, thyroarytenoid–lateral cricoarytenoid; CT, cricothyroid; SPL, sound-pressure level.

**Table 5 diagnostics-12-03124-t005:** Stepwise regression between LEMG and aerodynamics.

	β Value (95% Confidence Interval)			*p*-Value ^a^		
	Turn Ratio of TA-LCA	Turn Ratio of CT	Interaction between Turn Ratios of TA-LCA and CT	Turn Ratio of TA-LCA	Turn Ratio of CT	Interaction between Turn Ratios of TA-LCA and CT
Maximal SPL (dB)						
Mean SPL During Voicing (dB)						
Peak Air Pressure (cmH_2_O)						
Mean Peak Air Pressure (cmH_2_O)						
Peak Expiratory Airflow (L/s)	−0.34 (−0.65~−0.02)			0.036 *		
Mean Airflow During Voicing (L/s)	−0.28 (−0.50~−0.06)			0.014 *		
Aerodynamic Power (watt)	−0.42 (−0.78~−0.07)			0.019 *		
Aerodynamic Resistance (cmH_2_O)		14.93 (1.54~28.31)			0.029 *	
Aerodynamic Efficiency (ppm)	27.91 (6.24~49.59)			0.012 *		

^a^ Stepwise linear regression. * *p* < 0.05. LEMG, laryngeal electromyography; TA-LCA, thyroarytenoid–lateral cricoarytenoid; CT, cricothyroid; SPL, sound-pressure level.

## Data Availability

The datasets generated during and/or analyzed during the current study are available from the corresponding authors on reasonable request.

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
