# Peer review of "Aerodynamic Performance and Neuromuscular Control in Patients with Unilateral Vocal Fold Paralysis"

_diagnostics, 2022, doi:10.3390/diagnostics12123124_

Round 1
Reviewer 1 Report
The authors prospectively studied the relationship between quantitative LEMG and aerodynamic parameters in case of 134 unilateral vocal fold paralysis.
Introduction was very clearly expressed.
Material and methods chapter :
The technical description of the concentric needle is not mentioned.
Results:
In Table 1 legend, you don’t make the difference between numbers (%) or mean (SD), which should be explained.
In this Table 1, you showed the time post paralysis, which is very dispersed, allowing some EMG activity recovery, in some patients.
In legend of Table 2, ** is mentioned as <0.01. But <0.001, should be reported as ***, as usual.
No other comment.
Discussion :
“Translarygeal air flow” has to be replaced by Translaryngeal air flow in lines 309 & 310.
References :
Some references are the same : double references. They have to be suppressed.
For instance: 4 & 3; 24 & 12.
Reviewer 2 Report
The submitted manuscript reports a clinical study in which LEMG parameters of 134 patients with UVFP were related to aerodynamic parameters. The considered LEMG parameters of the TA-LCA muscle complex and the CT muscle were the peak turn frequency and the turn ratio. The considered aerodynamic parameters were the aerodynamic power, the aerodynamic resistance, the aerodynamic efficiency and the acoustic power. Those parameters were obtained using the Phonatory aerodynamic system KayPENTAX Model 6600. Statistical analysis was based on hypothesis testing, correlation analysis, and regression analysis. The study appears to be sound, given the number of included subjects and the rigour analyses.
My comments relate mainly to issues with the presentation style, and a few methodological recommendations are also made. The detailed comments are the following:
1. Regarding presentation style: There is no single figure in the manuscript, but a lot of table that are comparatively difficult to understand. In particular, I recommend adding the following figures:
a. An anatomic/functional sketch that explains the laryx physiology (esp. relevant nerves and muscles)
b. Visualisation of results (e.g., boxplots, scatter plots etc.) that support the main conclusions, instead of putting all results into tables
c. A figure illustrating what a “turn” is
d. A figure of the KayPentax device, acompanied by a description of its rational using a mask and not (only?) a microphone
2. The organization/formatting of the tables is often confusing. E.g. Table 1: some of the numbers in the brackets are standard deviations, others are proportions in %. I recommend not to mix those in one table without appropriate compartments. Also, the left column should not just be a list of words but reflect there hierarchy. Age is reported with respect to sex, counts of patients are reported with respect to paralysed side, time post paralysis is reported as a single average value (why?). Than a large portion of types of pathogenesis. Much is mixed in table 1, without a clear indication of the structure of the data.
3. Whenever possible, I recommend to follow STARD reporting guidelines : Cohen JF, Korevaar DA, Altman DG, Bruns DE, Gatsonis CA, Hooft L, Irwig L, Levine D, Reitsma JB, de Vet HCW, Bossuyt PMM. STARD 2015 guidelines for reporting diagnostic accuracy studies: explanation and elaboration. BMJ Open 2016;6:e012799
4. Diagnostic studies normally contain ROC curves and reports regarding sensitivity/specificity or similar.
5. A few key references appear to be missing, e.g., https://pubmed.ncbi.nlm.nih.gov/22576246/
6. I do not really understand the conclusions: how do the result support statements regarding the residual muscle function? Please explain. “Further impeding aerodynamics” Why further? Compared to what?
Limitations: very short listing of the two limitations mentioned by the authors. A bit more could be explained here. Also, the last sentence of this sections on limitations may be better put into the conclusions. By the way: I’d be interested to see a more specific description how the correlations reported in the study may provide a basis of therapeutic applications. I do not really see a connection there: What therapeutic applications? Voice therapy, Thyroplasty, electrical solutions? Please elaborate
Minor comments:
Line 207, why is it surprising that the peak turn frequency in the CT muscle was lower in the paralyzed side? Maybe I misunderstood the basics here, a better explanation would be appreciated.
LMEG vs. LEMG. Some typos across manuscript
